# Adaptive integrated intervention approaches for schistosomiasis elimination in Pemba: A 4-year intervention study and focus on hotspots

Lydia Trippler[1,2], Said Mohammed Ali[3], Mohammed Nassor Ali[3], Ulfat Amour Mohammed[3], Khamis Rashid Suleiman[3], Naomi Chi Ndum[1,2], Saleh Juma[4], Shaali Makame Ame[5], Fatma Kabole[5], Jan Hattendorf[1,2], Stefanie Knopp[1,2]*

1 Swiss Tropical and Public Health Institute, Allschwil, Switzerland, 2 University of Basel, Basel, Switzerland, 3 Public Health Laboratory – Ivo de Carneri, Chake Chake, Pemba, United Republic of Tanzania, 4 Neglected Diseases Program, Zanzibar Ministry of Health, Chake Chake, Pemba, United Republic of Tanzania, 5 Neglected Diseases Program, Zanzibar Ministry of Health, Zanzibar, Unguja, United Republic of Tanzania

* s.knopp@swisstph.ch

## Abstract

### Background

Schistosomiasis is a disease of poverty. Integrated interventions are recommended for its elimination. Despite major prevalence reductions over the past decades, hotspot areas with persistent or recurring moderate or high prevalence remain. We aimed to assess the contribution of multidisciplinary interventions that were adapted to the local micro-epidemiology for schistosomiasis elimination in Pemba, Tanzania, and to identify drivers for the occurrence of hotspot areas.

### Methodology

From 2020 to 2024, annual cross-sectional surveys were conducted in schools and communities in 20 implementation units (IUs) to assess the *Schistosoma haematobium* prevalence and monitor the impact of interventions. Based on the prevalence, the IUs were annually re-stratified into hotspot and low-prevalence IUs. In hotspots, mass drug administration in schools and communities, snail control and behavior change measures were implemented. Low-prevalence areas received surveillance-response interventions. With a random effects model, the association between *S. haematobium* infections and environmental and economic factors were assessed. Using risk layers based on the random effects model, hotspot areas were determined geographically.

### Principal findings

The overall *S. haematobium* prevalence in the 20 IUs changed from 1.2% (26/2200, 95% Confidence Interval (CI): 0.5-1.9%) in 2021 to 1.0% (27/2752, 95% CI: 0.4-1.6%)

**Data availability statement:** All relevant data are within the manuscript and its Supporting Information files.

**Funding:** Funding for the study has been obtained from the Swiss National Science Foundation (SNSF; Bern, Switzerland) via a PRIMA grant (PR00P3_179753 / 1) to SK. The funders had no role in study design, data collection and analysis, decision to publish, or preparation of the manuscript.

**Competing interests:** The authors have declared that no competing interests exist.

in 2024 in schools, and from 0.8% (31/3885, 95% CI: 0.4-1.2%) in 2021 to 1.2% (43/3711, 95% CI: 0.3-2.0%) in 2024 in communities. Across the study period, 8 IUs were considered a hotspot. The number of hotspot IUs decreased from 5 in 2021, to 4 in 2022, to 3 in 2023 but increased again to 5 in 2024. Some of the hotspot IUs resurged once interventions were adapted to surveillance-response. *S. haematobium* infections were significantly associated with the standardized kernel density of water bodies with *Bulinus* presence (Odds Ratio (OR): 2.3; 95% CI: 1.6-3.4), a very low economic score (OR: 4.1; 95% CI: 1.7-9.9) and living far away from a road (OR: 4.7; 95% CI: 2.1-10.6).

## Conclusion

Adaptive multidisciplinary interventions maintained the very low prevalence in Pemba but failed to interrupt *S. haematobium* transmission within 4 years. A comprehensive integrated intervention package contributed to reducing the number of hotspot IUs. However, some hotspots persisted also intense interventions or resurged once interventions were adapted to surveillance-response. To achieve complete elimination in Pemba and elsewhere, poverty needs to be reduced, and investments in global health equity, including the water sanitation and hygiene infrastructure, are essential.

## Trial registration

ISRCTN, ISCRCTN91431493. Registered 11 February 2020, https://www.isrctn.com/ISRCTN91431493.

## Author summary

Schistosomiasis is a disease of poverty and mainly affects deprived populations in sub-Saharan Africa. Efforts to control schistosomiasis have greatly reduced its prevalence over the past decades, but some hotspots of transmission remain. To achieve elimination, integrated interventions are recommended by the World Health Organization. We aimed to assess the contribution of multidisciplinary interventions that were adapted to the local micro-epidemiology for schistosomiasis elimination in Pemba, Tanzania, and to identify drivers for the occurrence of hotspots. Based on the results of annual surveys, we stratified our study area into "hotspot" and "low-prevalence" areas. In hotspots, interventions included the mass deworming of people, mollusciciding against the intermediate host snails, and health communication to induce a behavior change. In low-prevalence areas, we implemented a surveillance-response system. Every year, the intervention approach was adapted to the current endemic situation. Over the 4 study years, the adaptive multidisciplinary interventions maintained the very low prevalence in Pemba but failed to interrupt *S. haematobium* transmission. The comprehensive integrated intervention package reduced the number of hotspots. However, some hotspots persisted or resurged once interventions were adapted to

surveillance-response. *Schistosoma haematobium* infections were particularly found in people residing near a water body containing intermediate host snails and in very rural areas and/or in households which had a very low economic score. To achieve complete elimination, poverty needs to be reduced and investments in global health equity, including the water sanitation and hygiene infrastructure, are essential.

## Background

Schistosomiasis is a disease of poverty that mainly affects poor and deprived populations [1]. It is endemic in 78 countries of the world, among which 41 are located in sub-Saharan Africa [2]. The global burden caused by schistosomiasis was estimated at 1.86 million disability adjusted life years (DALYs) in 2021 [3].

The World Health Organization (WHO) envisions the elimination of schistosomiasis as a public health problem globally and the interruption of *Schistosoma* transmission in humans in selected countries by 2030 [4, 5]. To help reduce transmission and achieve these goals, the WHO recommends applying mass drug administration (MDA) with praziquantel, water sanitation and hygiene (WASH) measures, environmental modifications including snail control, and behavior change interventions [4, 6]. Large-scale praziquantel MDA over the past two decades has resulted in a considerable decrease in the overall *Schistosoma* prevalence and morbidity in sub-Saharan Africa [7, 8]. Access to and use of WASH infrastructure are associated with a reduced risk of *Schistosoma* infection, but require a considerable amount of resources [9, 10]. Control of intermediate host snails is considered an important complementary measure to MDA, particularly in persistent hotspot areas, and as an essential tool to ultimately eliminate local transmission [6]. Once MDA and snail control are phasing out, environmental and behavioral factors are needed to prevent a rebound of infection [6]. In areas, where low prevalences have been achieved, MDA may be replaced by test-and-treat interventions [6].

Overall, countries have progressed well in the control and elimination of schistosomiasis [7, 11]. However, the focality of *Schistosoma* transmission and a pronounced spatial heterogeneity in prevalence levels are a challenge for elimination [8, 12, 13]. Several countries that are approaching elimination contain areas where prevalences are low and other areas where transmission and infection remain high, despite repeated interventions [13–15]. For a progress towards elimination by an adequate use of resources, it will be of utmost importance to consider the local micro-epidemiology and to target and adapt interventions accordingly [16–18]. While in low-prevalence areas, MDA may be stopped and replaced by test-and-treat or other individualized interventions, remaining hotspot areas constitute a particular challenge for reaching the elimination goals and hence require comprehensive intervention strategies.

In the SchistoBreak study implemented from 2020 to 2024, we assessed the contribution of three years of multidisciplinary intervention approaches that were targeted and adapted to the local micro-epidemiology, to urogenital schistosomiasis elimination in Pemba [19]. Here, we focused on the impact of all applied interventions on the overall prevalence in the study area and particularly on the impact of integrated MDA, snail control and behavior change interventions on the prevalence in hotspot areas. In addition, we identified factors that can help to explain the occurrence and persistence of hotspots.

## Methods

### Ethics

The SchistoBreak study protocol was waived by the ethics committee of Northwestern and Central Switzerland (EKNZ) on October 23, 2019 (Req-2019–00951). It subsequently received annual ethical approval by the Zanzibar Health Research Institute (ZAHRI). The first approval was given on December 13, 2019 (ZAHREC/03/PR/December/2019/12), and the latest approval was given on March 31, 2023 (ZAHREC/04/AMEND/MARCH/2023/03). The study was registered prospectively at ISRCTN (ISCRCTN91431493).

At the beginning of each project phase (annual surveys and intervention periods), approximately every six months, the leaders (shehas) of the implementation units (IUs) and all school principals were invited to the Public Health Laboratory-Ivo de Carneri (PHL-IdC) in Chake Chake, Pemba. In the meetings, updates on the SchistoBreak project results were presented. Moreover, challenges that occurred during the implementation of surveys of interventions were discussed, and the shehas and principals were invited to support the project by announcing the forthcoming activities in their communities (shehias) and schools and by motivating their population to participate.

All participants in the surveys and surveillance interventions were given information sheets about the project procedures, including the telephone number of the local study leader in case of any questions. Moreover, all participants were required to sign an informed consent form. In the case of participating children <18 years, their legal guardians were requested to sign the informed consent form. Additionally, children aged 12–17 years were asked to sign an assent form indicating their agreement to participate in the study.

## Study site

The SchistoBreak study was conducted on Pemba, an island in the Zanzibar archipelago of the United Republic of Tanzania, from 2020 to 2024. Pemba is divided into four districts, which contain 129 small administrative areas, called shehias [20]. The SchistoBreak study was conducted in the northern districts Wete and Micheweni, in a total of 20 shehias [19]. Each shehia in the study area was considered an IU. Based on the most recent Tanzanian population census, these 20 IUs had an estimated population size of approximately 95,000 people in 2022 [20]. In and around the SchistoBreak study area, 23 health facilities (21 primary health care units (PHCUs) and two hospitals) are located, providing basic and specialized healthcare services.

Schistosoma haematobium is the only autochthonous Schistosoma species infecting humans in Zanzibar [21, 22]. Historically, the S. haematobium prevalence in Pemba was very high and inflicted a considerable public health burden [23, 24]. However, stringent control and elimination interventions implemented across the last century greatly reduced the overall prevalence and morbidity [17, 22, 25, 26]. In 2020, the overall prevalence was 2.9% in school-aged children and 0.8% in adults [22, 27]. Yet, there is considerable spatial heterogeneity and while most areas show a zero or very low prevalence, some hotspot areas also exist [27].

## Study design and participants

The SchistoBreak study was designed as a 4-year intervention study, with annual parasitological surveys and interjacent intervention periods [19]. The four annual parasitological surveys employed a cross-sectional sampling approach in schools and communities. The multidisciplinary interventions were targeted to the local micro-epidemiology of S. haematobium, as described below. Sample size calculations are described in the published study protocol [19].
Eligible to participate in the surveys and interventions were all individuals who i) attended a school in the study area or lived in the study area, ii) were aged ≥4 years, and iii) provided a signed informed consent form for participation.

## Cross-sectional parasitological school-based surveys

At baseline in 2020/21 and subsequently, after each intervention period in 2021/22, 2022/23, and 2023/24, cross-sectional parasitological surveys were conducted in schools and households within the 20 IUs of the SchistoBreak study area. The surveys served i) to stratify the study area into hotspot and low-prevalence IUs to assign interventions in line with pre-defined prevalence thresholds and ii) to monitor the prevalence of S. haematobium infections and microhematuria over the study years [19]. The surveys were conducted annually between November and February/March.

For the school-based survey, the largest public primary school was selected in each IU. At baseline, 16 of the 20 IUs had a public primary school. In subsequent study years, 18 of the 20 IUs had a public primary school. In IUs without a public primary school, no school-based survey was conducted.

The school-based surveys were conducted in seven grades (nursery, grades 1–6) at each school. For each grade, a class was selected randomly for participation, based on a computer-generated list. For each grade, it was aimed to include 20 children. Accounting for a 20% dropout rate, 25 children were randomly selected per class. For randomization, children were asked to line up, sorted by sex. Subsequently, every third child from each line was selected for a total of 25 children. The study was explained to the children in Kiswahili, and they were given an information sheet along with informed consent and assent forms. Demographic data, such as age and sex, were collected by the study team. On the following day, after submitting the signed informed assent and consent forms, the children were asked to provide a urine sample between 10 am and 2 pm. The urine samples were stored in a cool box and transported to the PHL-IdC.

## Cross-sectional parasitological household-based surveys

A household-based cross-sectional survey was conducted in all 20 IUs in each study year. Aiming for 250 participants per IU, 70 housing structures per IU were randomly pre-selected in 2021, accounting for an average of five household members and a 30% drop-out rate [28, 29]. For the years 2022–2024, 80 housing structures were pre-selected accounting for a 40% drop-out rate based on experiences from the baseline survey [29]. For the selection of the housing structures, centroids of spatial polygons of buildings in Pemba were extracted from shape files, provided by the Zanzibar Commission for Lands to the Zanzibar Neglected Tropical Diseases (NTD) Program, using ESRI ArcMap 10.6.1 [28]. Subsequently, the centroids to be included into the survey were randomly selected using R (www.r-project.org). With the combination of a mobile application for data collection (Open Data Kit (ODK), www.opendatakit.org) and a mobile application for offline navigation (Maps.Me, www.de.maps.me, in 2021 and 2022 and OrganicMaps, www.organicmaps.app, in 2023 and 2024), both installed on Samsung Galaxy A Tabs, the centroids of the housing structures were locatable by the study team in the communities.

Each IU was visited by the study team for three consecutive days. On day one, the housing structures were located. It was assessed whether the housing structures were residential houses or other housing structures, such as shops, mosques, and schools. In residential houses, the field enumerators explained the purpose of the study to present household members in Kiswahili. All eligible household members were invited to participate in the survey by signing the informed consent and assent forms and providing a fresh urine sample. A questionnaire to collect demographic data of all household members and additional information was conducted with one present adult household member. In 2022–2024, household-economic data were collected, such as the type of roof and floor, and possessions, such as a TV, radio, bicycle, or access to electricity at home. If some household members were not present during the first visit, urine cups were left with a present household member, marked with unique participant identifier codes and stickers with drawings to identify the intended recipient. The present household members were instructed to ask the returning household members to provide a urine sample and signed informed consent and assent forms the following morning.

On the second and third day, the study team revisited all households where they had previously distributed urine cups and consent forms to collect any remaining items.

## Classification into hotspot and low-prevalence implementation units

To target multidisciplinary intervention strategies for urogenital schistosomiasis elimination to the local micro-epidemiology of *S. haematobium* in Pemba, the 20 IUs were classified into hotspot and low-prevalence IUs. The selection of thresholds used to distinguish a "hotspot" IU from a low-prevalence IU in this study was based on age-dependent infection rates [19]. Every IU with a prevalence of ≥3% in the school-based survey and/or a prevalence of ≥2% in the household-based cross-sectional survey was classified as a hotspot IU. Every IU with a prevalence of <3% and <2% in the school-based and household-based survey, respectively, was classified as a low-prevalence IU. The prevalence estimates for the thresholds were based on the results of a single urine filtration. The classification of hotspot and low-prevalence IUs and respective intervention approaches was renewed annually, in line with survey results. Of note, the thresholds were set at

such a low prevalence levels since i) Pemba is an elimination setting with a very low overall prevalence [25, 27] and ii) this study was the first study conducted in Pemba that stopped MDA in designated low-prevalence IUs to evaluate a surveillance-response approach including a test-treat-test-track-test-treat (5T) intervention [19, 30]. We hence aimed to minimize the risk of recrudescence and respond rapidly to any increase in prevalence in IUs.

### Interventions in hotspot implementation units

In all hotspot IUs, multidisciplinary interventions were implemented. Those included (bi-)annual MDA with praziquantel, snail control, and behavior change interventions.

School- and community-based MDAs were conducted at least annually by the Pemba NTD Program of the Zanzibar Ministry of Health and aimed to reach all inhabitants of Pemba aged ≥4 years. During school-based MDA, children were provided with porridge in the morning to increase bioavailability and to reduce potential adverse events before they were treated by teachers with praziquantel (40 mg/kg single oral dose) using a dose pole [31] who were supervised by a member of the NTD Program. During community-based MDA, experienced drug distributors went from house to house, providing praziquantel to community members who had not received praziquantel in school-based MDA.

Snail control was implemented at all known freshwater bodies in the hotspot IUs where intermediate host snails of the genus *Bulinus* were found [19]. During each intervention period, the waterbodies were visited four times, and snail surveys were conducted. For this purpose, two experienced field workers entered the shorelines of the waterbodies and searched for freshwater snails across a 20-meter distance for ten minutes. If freshwater snails were found, the genus was recorded. If *Bulinus* was found, the number was counted and recorded and snails were taken to PHL-IdC for examination. Moreover, if *Bulinus* were found on the day of the survey or if there was a history of *Bulinus* in the respective waterbody (i.e., when *Bulinus* had been found previously during the Zanzibar Elimination for Schistosomiasis Transmission (ZEST) or Schisto-Break study), the molluscicide niclosamide (WP83,1; Bayer AG Crop Science Division, Monheim, Germany) was applied for snail control to reduce the number of intermediate host snails for schistosomiasis [32]. Depending on the size and nature of the water body, niclosamide was sprayed using plastic backpack sprayers (Farmate, Taizhou Sunny Agricultural Machinery Co., LTd, Taizhou, China) or a gasoline-powered sprayer (Zhejiang O O Power Machinery Co., Ltd, Zhejiang, China) at an initial concentration of 8–10 mg solved in one liter water taken from the water body.

Behavior change interventions were implemented in schools and communities in the hotspot IUs, based on a human-centered design approach that was developed during the ZEST Study [33]. In each IU that was defined a hotspot during the SchistoBreak study years, two washing platforms were installed near a pump, tap or well to provide a convenient alternative for household chores such as washing clothes or dishes, instead of using water from natural open water bodies [34]. The locations where washing platforms were placed were identified by community members and community leaders in close collaboration with the SchistoBreak study team. Community members, supported by the study team, built the platforms.

In all hotspot communities, community meetings were held to educate community members about interventions and preventive measures against schistosomiasis. Furthermore, community members were informed about the construction of the washing platforms and encouraged to use them.

Behavior change interventions were also implemented in primary, secondary, and Islamic schools. During each intervention period, teachers of the schools in the hotspot IUs were invited to a meeting and training at PHL-IdC. Here, the teachers learned about interactive teaching methods, including the drawing of *S. haematobium* life cycle pictures, the use of a blood fluke picture, and the application of self-made snail boards to educate their schoolchildren about urogenital schistosomiasis. Moreover, the teachers were trained in safe games with educational messages about schistosomiasis and the preparation for school outreach days (Kichocho Days). Schools participating in the behavior change activities were regularly visited by the SchistoBreak study team who supported the teachers and children in the preparation and conduction of the Kichocho Days. During the Kichocho Day, children and other participating individuals such as teachers

or community members were educated about the transmission of *S. haematobium*, its symptoms, and preventive measures by a member of the study team. Moreover, children performed songs, poems, and dramas and were engaged in games that could serve as an alternative to playing in water bodies. All components of a Kichocho Day contained educational messages about schistosomiasis that were transmitted to a wide audience in a playful and elucidating manner.

### Interventions in low-prevalence implementation units

In all low-prevalence IUs, surveillance-response interventions were implemented [19, 30]. As part of active surveillance-response activities, all children in grades 3–5 in the largest public primary school and one Islamic school were tested for a *S. haematobium* infection and microhematuria once they had submitted their informed consent form signed by their parent or legal guardian. If a child tested positive for *S. haematobium* or microhematuria, the child was treated with praziquantel (40 mg/kg body weight) using a dose pole by a study team member. Moreover, positive children were accompanied to their homes and to the water bodies they used. For reactive surveillance-response, household members present at home and individuals present at the waterbodies were invited for testing and praziquantel treatment if positive. The indicated water bodies were also surveyed for intermediate host snails as described above for hotspot IUs. If *Bulinus* were present or had been found in the same water body during earlier surveys of the SchistoBreak study, niclosamide was applied for snail control.

### Passive surveillance interventions in hotspot and low-prevalence implementation units

Collaborations for the implementation of passive surveillance interventions including all health facilities across the SchistoBreak study area were started in 2021. Once (2021 and 2023) or twice (2022) a year, two employees from each PHCU and three employees from each hospital were invited to attend a meeting at PHL-IdC. During these meetings, health facilities employees were educated about urogenital schistosomiasis transmission and prevention and symptoms related to urogenital schistosomiasis [35]. They also received updates on the SchistoBreak study and were trained in the use of reagent strips (Hemastix) to test for microhematuria as a proxy for *S. haematobium* infections. Moreover, the needs of the health facilities and staff regarding diagnosis and reporting for the study were discussed. At the end of each meeting, the health facilities were equipped with Hemastix, praziquantel, and paper forms to record patients with schistosomiasis-related symptoms and whether they tested microhematuria-positive. The health facility staff were asked to complete the paper forms regularly, and a member of the SchistoBreak study team collected the forms every other week.

### Laboratory procedures

All urine samples collected during the cross-sectional school- and household-based surveys and during surveillance activities in the intervention periods, were examined for *S. haematobium* eggs using the urine filtration method and for microhematuria as a proxy for a *S. haematobium* infection using Hemastix reagent strips [36].

For the examination of *S. haematobium* infections with the urine filtration method, all urine samples collected in parasitological surveys and surveillance activities were brought to the laboratory at PHL-IdC. Here, 10 ml of each urine sample were filtered through a 13 mm fabric filter (Sefar Ltd., Bury, United Kingdom), held by a Swinnex plastic filter holder (Millipore, Merck KGaA, Darmstadt, Germany), using a 10 ml plastic syringe. Subsequently, the filter was removed from the holder and examined under a light microscope to determine the presence and quantity of *S. haematobium* eggs. Samples containing 1–49 eggs per 10 ml of urine were classified as light-intensity infection, while those containing 50 or more eggs per 10 ml of urine were classified as heavy-intensity infection [37].

In addition, urine samples were examined for microhematuria using reagent strips (Hemastix; Siemens Healthcare Diagnostics AG; Zürich, Switzerland). The examination results were graded as microhematuria negative, or as trace, small (+), moderate (++), or large (+++) microhematuria, according to the manufacturer's color chart. During the annual parasitological surveys, microhematuria was assessed in the laboratory of PHL-IdC. During the surveillance activities, microhematuria was assessed at the point of contact in the schools, households, or water bodies.

## Data management

Data collected during questionnaire interviews in the cross-sectional surveys and the intervention periods, data on snail presence and on water body characteristics, and data on implemented behavior change interventions were recorded using ODK, installed on Samsung Galaxy Tab A tablets. The results of the urine filtration examination from the 2021 surveillance-response intervention were recorded using ODK. Data retrieved from paper forms filled by health facilities during the passive surveillance were entered into ODK. Data collected using ODK were sent to a secured ODK server hosted at the Swiss Tropical and Public Health Institute in Allschwil, Switzerland.

The laboratory results of all cross-sectional surveys and of the surveillance-response interventions conducted in 2022 and 2023 were recorded on paper in the laboratory and then double-entered into a Microsoft Excel spreadsheet (version 2016) by two experienced data entry clerks. Any discrepancies in the double-entered data were verified by checking results entered in the original paper forms and then corrected electronically.

The laboratory results were merged with the questionnaire and registration data to perform statistical analysis of the data collected during cross-sectional surveys and the intervention periods. To inform participants of their *S. haematobium* infection status, their names were linked with the laboratory analysis data. Otherwise, names were kept in a separate file, and only coded data were used for statistical analyses. All data cleaning was conducted using R and STATA/IC 16.1.

## Statistical analysis

To assess the household-economic status of individuals participating in the household-based surveys, a principal component analysis (PCA) was conducted based on 13 variables indicating wealth (type of floor, type of toilet, type of cooking material, type of roof, type of drinking water, electricity availability, radio availability, refrigerator availability, television availability, mobile phone availability, bicycle availability, motorbike availability, car availability). Individuals with missing values in at least one of the 13 variables were excluded. The 13 variables were standardized to a mean of zero and a standard deviation of 1 to ensure that each variable contributed equally to the PCA. The PCA was conducted on the selected variables and the contribution of each variable to the principal components was evaluated by printing the loadings.

To determine the chance of being infected with *S. haematobium*, a random effects model was conducted with *S. haematobium* infection in the household-based survey as the dependent variable. The independent variables included demographic information (sex and age) and the first principal component indicating the household economic status, which was grouped through natural breaks. Further independent variables were the density of water bodies with *Bulinus* presence, standardized to a mean of zero and a standard deviation of 1, and the distance from the household to the next road as a proxy for the rural/urban location of the household. The IU was included in the model as the random effect. Odds ratios (ORs) with 95% confidence intervals (CIs) with a statistical significance at the 5% level were assessed. Since no economic data were collected in the household-based survey 2021, only complete data from 2022, 2023, and 2024 were included. The road data were retrieved from the Humanitarian Data Exchange Platform (https://data.humdata.org/dataset/hotosm_tza_roads), published under the Open Database License (ODC-ODbL).

Based on the odds obtained by the random effects model, spatial risk layers for the kernel density of water bodies with *Bulinus*, density of different household-economic values, and distance to road were created. These risk layers were subsequently combined into a risk map, indicating areas with a low to high chance of infection.

## Results

### Study flow and participant characteristics

In the four school-based cross-sectional surveys from 2021 to 2024, a total of 11614 children from 18 schools (16 in 2021) were randomly selected (Fig 1). Of the 11614 registered children, two (0.02%) children were excluded from the analysis due to non-eligibility (<4 years old), 1385 (12.0%) children were excluded due to their absence on the day of urine

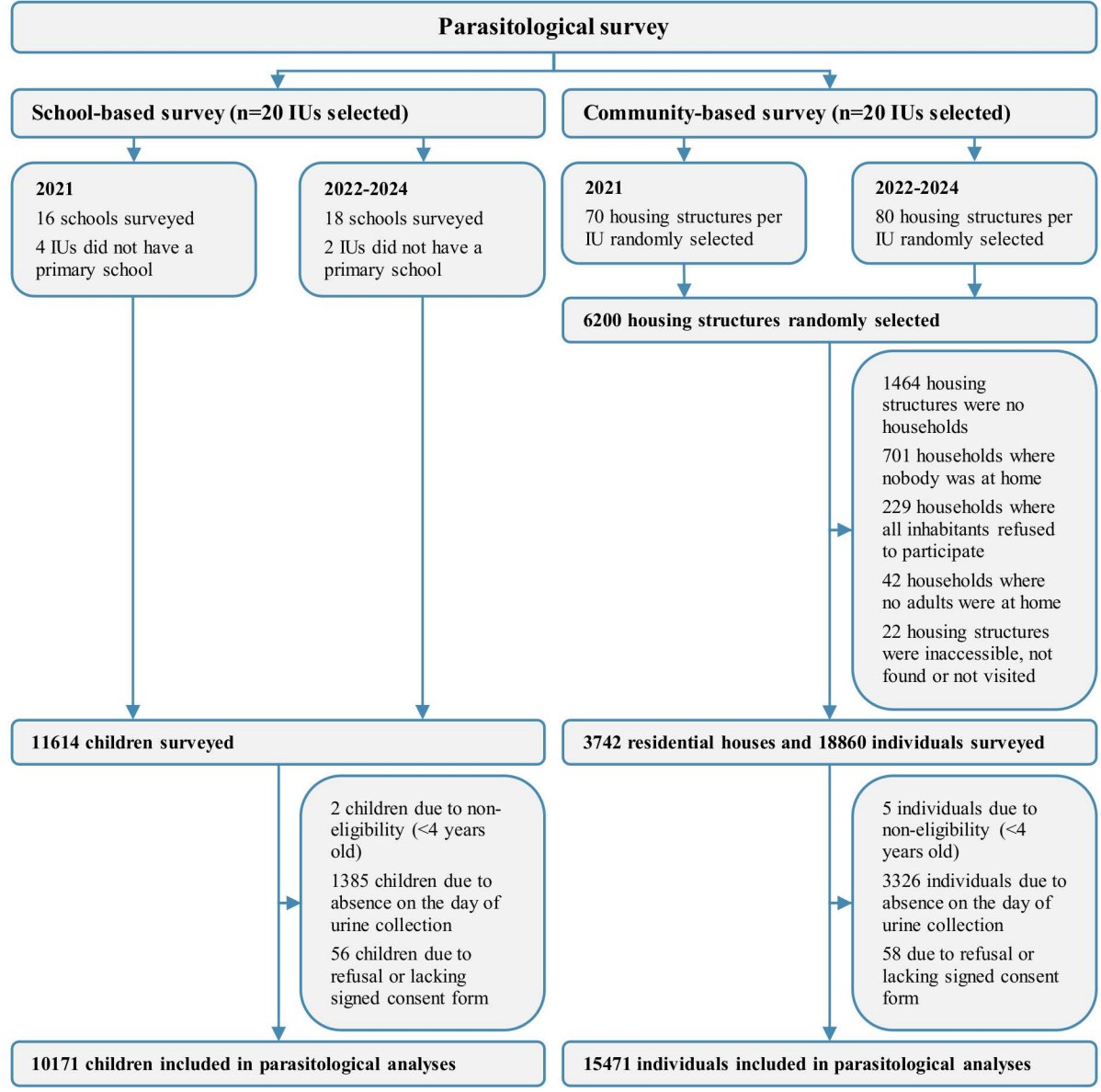

**Fig 1. Study participation.** Flow diagram of individuals participating in the cross-sectional school- and household-based surveys of the SchistoBreak study in Pemba, Tanzania, from 2021 to 2024. IU = implementation unit.

collection and 56 (0.5%) were excluded due to refusal or not submitting a signed consent form. Finally, urine samples from 10171 children were collected and included in the parasitological analysis.

In the household-based surveys from 2021 to 2024, a total of 6200 housing structures were randomly selected in the 20 IUs (Fig 1). Of those, 1464 (23.6%) were no households, in 701 (11.3%) households, nobody was at home at the time of visit, in 229 (3.7%) households, all individuals refused to participate, in 42 (0.7%) households, no adult person was at home at the time of visit and 22 (0.4%) housing structures were inaccessible, were not found or not visited. Hence, in total, 3742 residential houses and 18860 individuals were surveyed. Of the 18860 individuals, 5 (0.03%) were excluded

due to non-eligibility (<4 years old), 3326 (17.6%) were excluded due to absence on the day of urine collection or since they did not submit a urine sample, and 58 (0.3%) individuals refused to participate or did not submit an informed consent form. Finally, urine samples from 15471 individuals were collected and included in the parasitological analyses.

Of the 10171 children participating in the school-based surveys that were included in the analysis, 51.0% (5183/10171) were female, and 49.0% (4988/10171) were male (Table 1). The median age of the participating children was 9 (4–17) years. Of the 15471 individuals participating in the household-based survey included in the analysis, 55.0% (8503/15471) were female, and 45.0% (6968/15471) were male. The median age of the participating individuals was 17 (4–102) years.

### *Schistosoma haematobium* and microhematuria prevalence from 2021 to 2024

The results of the baseline school-based survey conducted in 2021 showed that 1.2% (26/2200) of the participating children were egg-positive for *S. haematobium*, with 0.2% (4/2200) of the participants having a heavy-intensity infection (Fig 2A). After one year of interventions in hotspot and low-prevalence areas, the overall prevalence changed to 0.9% (22/2527) in 2022, with 0.1% (3/2527) heavy-intensity infections. In 2023, 1.0% (27/2684) and 0.3% (9/2684) of the children were infected with *S. haematobium* and had heavy-intensity infections, respectively. In the final survey in 2024, 1.0% (27/2752) of the children tested egg-positive for *S. haematobium*, with 0.1% (3/2752) having a heavy-intensity infection.

The baseline household-based survey conducted in 2021 revealed a *S. haematobium* prevalence of 0.8% (31/3885) (Fig 2B). Among the individuals tested, 0.1% (3/3885) had a heavy-intensity infection. After one year of interventions, in 2022, 0.9% (34/3963) of the participants tested egg-positive, with 0.2% (7/3963) heavy-intensity infections. In 2023, the prevalence changed to 1.0% (40/3844), with 0.1% (5/3844) heavy-intensity infections. In the final year of the SchistoBreak study, 1.2% (43/3711) of the participants were egg-positive, and 0.1% (5/3711) had a heavy-intensity infection.

The results of the baseline school-based survey conducted in 2021 showed that 3.5% (78/2202) of the participants were microhematuria-positive, with 0.7% (15/2202) having large microhematuria (Fig 2A). After one year of interventions in hotspot and low-prevalence areas in 2022, the overall microhematuria prevalence changed to 6.4% (163/2529), with 0.9% (23/2529) of children having large microhematuria. In 2023, 5.4% (145/2680) of the children tested microhematuria-positive, with 0.6% (17/2680) having large microhematuria. In the final survey of the SchistoBreak study conducted in 2024, 2.7% (73/2752) of the children were microhematuria-positive, with 0.4% (11/2752) having large microhematuria.

**Table 1. Demographic information of study participants.** Demographic information of participants in school-based and household-based cross-sectional surveys of the SchistoBreak study implemented in Pemba, Tanzania, from 2021 to 2024.

| | | School-based survey | | | | Household-based survey | | | |
|---|---|---|---|---|---|---|---|---|---|
| | Total (N = 25642) | 2021 (N = 2202) | 2022 (N = 2532) | 2023 (N = 2684) | 2024 (N = 2753) | 2021 (N = 3895) | 2022 (N = 3985) | 2023 (N = 3848) | 2024 (N = 3743) |
| **Sex** | | | | | | | | | |
| Female | 13686 (53.4%) | 1170 (53.1%) | 1291 (51.0%) | 1350 (50.3%) | 1372 (49.8%) | 2068 (53.1%) | 2203 (55.3%) | 2116 (55.0%) | 2116 (56.5%) |
| Male | 11956 (46.6%) | 1032 (46.9%) | 1241 (49.0%) | 1334 (49.7%) | 1381 (50.2%) | 1827 (46.9%) | 1782 (44.7%) | 1732 (45.0%) | 1627 (43.5%) |
| **Age (years)** | | | | | | | | | |
| Median [Min, Max] | 12.0 [4.00, 102] | 10.0 [4.00, 17.0] | 9.00 [4.00, 16.0] | 9.00 [5.00, 16.0] | 9.00 [5.00, 16.0] | 17.0 [4.00, 87.0] | 18.0 [4.00, 97.0] | 18.0 [4.00, 92.0] | 16.0 [4.00, 102] |
| Missing | 3 (0%) | 2 (0.1%) | 0 (0%) | 0 (0%) | 0 (0%) | 1 (0%) | 0 (0%) | 0 (0%) | 0 (0%) |

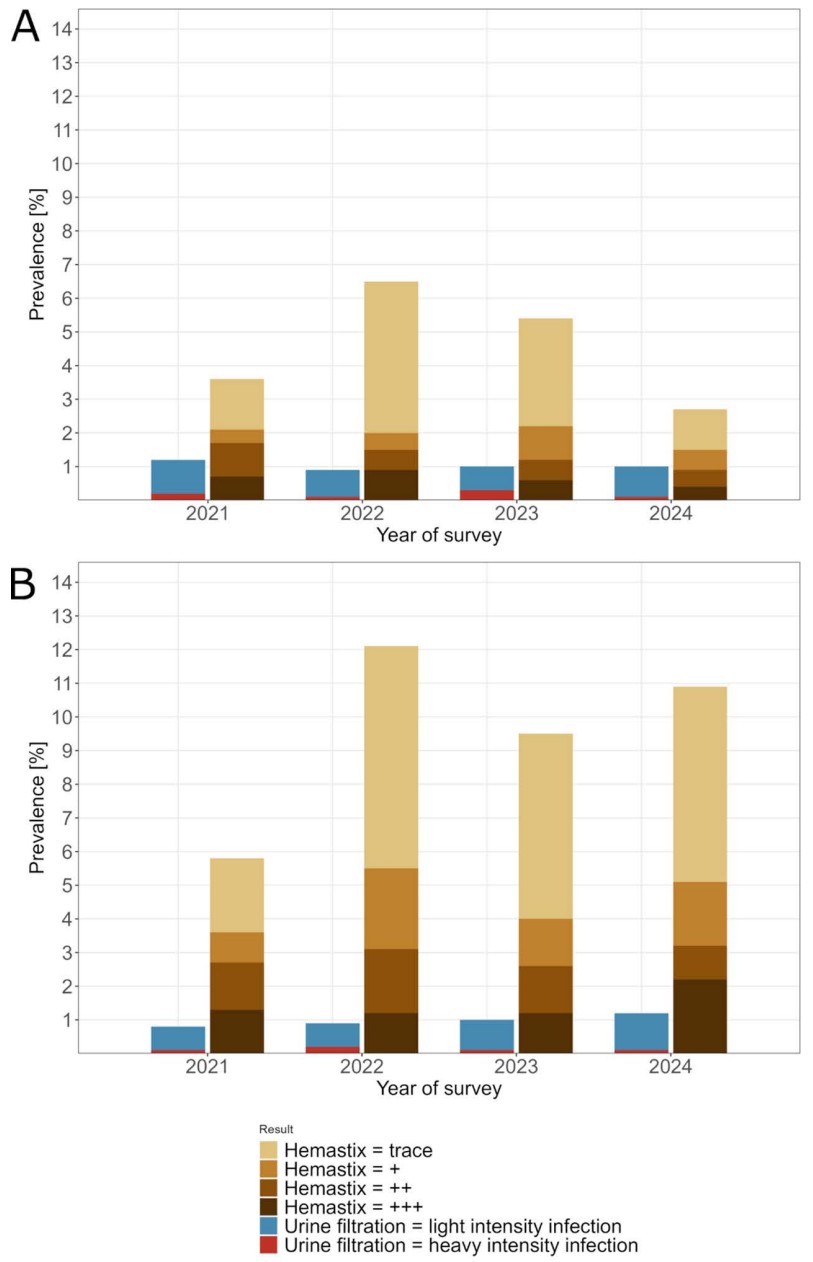

**Fig 2. *Schistosoma haematobium* and microhematuria prevalence.** *Schistosoma haematobium* and microhematuria prevalence among schoolchildren (A) and household members (B) in annual parasitological surveys in the SchistoBreak study area in Pemba, Tanzania, from 2021 to 2024.

The baseline household-based survey conducted in 2021 revealed a microhematuria prevalence of 5.8% (225/3885) among the participants (Fig 2B). Out of the 3885 individuals tested, 1.3% (49/3885) had large microhematuria. After one year of interventions, in 2022, the microhematuria prevalence was 12.0% (472/3963) with 1.2% (47/3963) large microhematuria. In 2023, 9.4% (359/3844) of the participants tested microhematuria-positive, with 1.1% (44/3844) large microhematuria. In the final year of the SchistoBreak study, the microhematuria prevalence was 10.9% (404/3711) with 2.2% (83/3711) large microhematuria.

## Annual changes in the number of hotspot and low-prevalence implementation units

In the baseline cross-sectional parasitological survey in 2021, three schools and three communities had a *S. haematobium* prevalence of ≥3% and ≥2%, respectively, which resulted in a total of 5 hotspot IUs and 15 low-prevalence IUs (Fig 3). After the first intervention period, one hotspot IU remained a hotspot, four hotspot IUs turned into low-prevalence IUs, 12 low-prevalence IUs remained low-prevalence and three turned into hotspots. Hence, in line with the results of the cross-sectional parasitological survey conducted in 2022, four IUs were hotspots and 16 IUs were low-prevalence IUs, which received respective interventions in the subsequent period. Following the same procedure, three IUs were identified as hotspots and 17 were classified as low-prevalence areas in 2023. In the final survey in 2024, five IUs had crossed the prevalence threshold for hotspots and 15 were categorized as low-prevalence IUs.

Throughout three intervention periods, twelve IUs (60%) never were a hotspot IU, four IUs (20%) were a hotspot once, and four IUs (20%) were a hotspot IU in two intervention periods. Of the latter four IUs, one IU was considered a hotspot for the first two intervention periods, fell then below the prevalence threshold, and received the low-prevalence

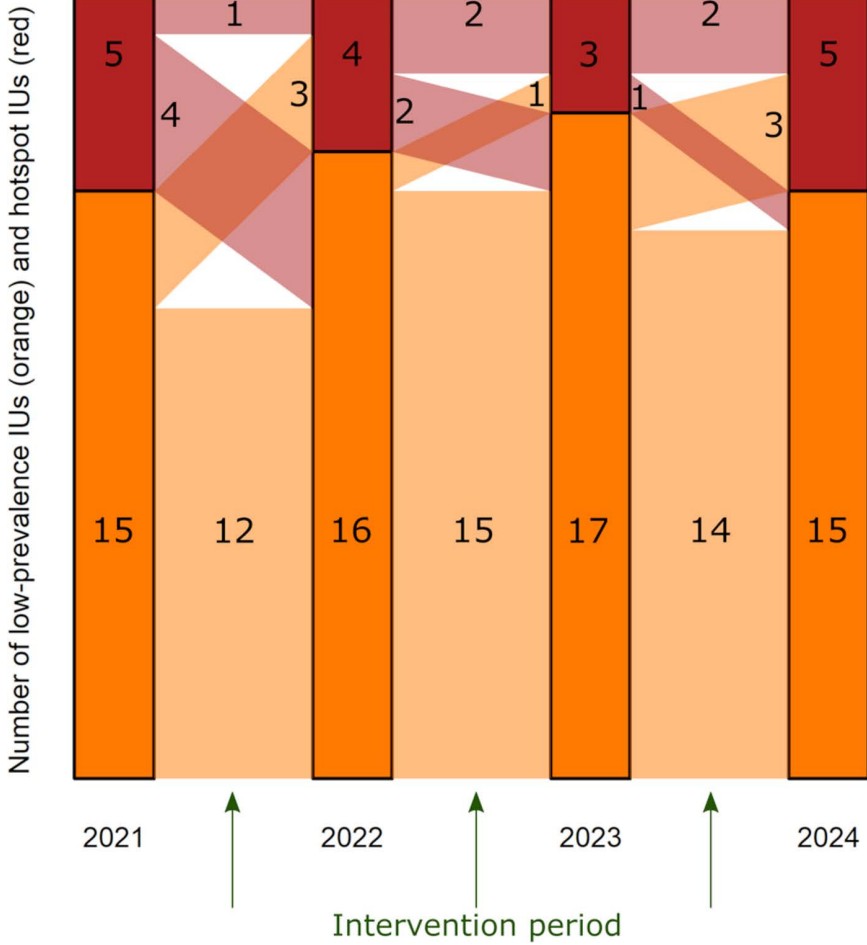

**Fig 3. Annual change in low-prevalence (orange) and hotspot (red) implementation units (IUs).** Changing number of IUs classified as low-prevalence IUs and hotspot IUs through annual cross-sectional surveys in Pemba, Tanzania, from 2021 to 2024. Green arrows indicate the intervention periods between the annual cross-sectional surveys with multidisciplinary interventions in hotspot areas and surveillance-response interventions in low-prevalence areas.

interventions in the third intervention period. The second IU was considered a hotspot in the first intervention period and became a low-prevalence IU after one year of intense interventions. Subsequently, after receiving the surveillance-response interventions in the second intervention period, the IU bounced back, and was considered a hotspot IU in the third intervention period. The third and fourth IUs were considered low-prevalence IUs in the first intervention period and received surveillance-response interventions, but became and remained a hotspot IU in the following two intervention periods. At the final survey in 2024, two of the 8 hotspot IUs remained above the prevalence threshold, one hotspot IU decreased to low-prevalence levels, one IU that had received surveillance-response interventions during the last intervention period increased to a hotspot level, and four IUs that had received surveillance-response interventions remained low-prevalence IUs.

## Multidisciplinary interventions in hotspot areas

In the annual intervention periods 2021–2023, a total of four MDA rounds were conducted in the hotspot IUs (Table 2). Across the three intervention periods, a total of 112 different water bodies were surveyed for *Bulinus* in the 8 hotspot IUs. In 44 (39.3%) among the 112 water bodies, *Bulinus* were found in at least one survey, and 50 (44.6%) of the 112 waterbodies were treated with niclosamide for snail control, either because *Bulinus* were found during the current or past surveys. In total, 654 malacological surveys were conducted and snail control was conducted 304 times.

In the 8 IUs that were considered a hotspot, a total of 16 washing platforms were constructed, and 112 Kichocho Days, 49 community meetings, and 6 teacher trainings were conducted for behavior change.

## *Schistosoma haematobium* prevalence in hotspot implementation units

In 2021, 2.8% (19/669) of the children participating in the school-based survey in the five IUs considered hotspots tested egg-positive for *S. haematobium* (Fig 4A), and 4.6% (31/669) of the children tested positive for microhematuria (S1 Text and S1A Fig). Of the 669 children participating, two children (0.3%) were diagnosed with a heavy-intensity infection. After

**Table 2. Interventions in the hotspot implementation units (IUs). Interventions in the hotspot IUs of the SchistoBreak study in three intervention periods in Pemba, Tanzania, from 2021 to 2023.**

|  | 2021 | 2022 | 2023 |
|---|---|---|---|
| Hotspot IUs | 5 | 4 | 3 |
| **MDA** |  |  |  |
| MDA rounds | 1 | 1 | 2 |
| Coverage in schools[a] | 94.4% | 96.7%* | 87.1%** |
| Coverage in communities[a] | 70.8% | 85.3%* | 84.5%** |
| **Snail control** |  |  |  |
| Water bodies surveyed | 50 | 59 | 57 |
| Water bodies treated with niclosamide when *Bulinus* spp was found in the present or past | 30 | 19 | 33 |
| Surveys at water bodies | 196 | 235 | 223 |
| Molluscicide treatments of water bodies | 98 | 73 | 133 |
| **Behavior change** |  |  |  |
| Washing platforms constructed | 10 | 6 | 0 |
| Kichocho Days | 30 | 44 | 38 |
| Community meetings | 16 | 30 | 3 |
| Teacher training | 2 | 2 | 2 |

[a]Coverage data were provided by the NTD program of the Zanzibar Ministry of Health, *Coverage data were available for one among four hotspot IUs only, **Coverage data were available for two among three hotspot IUs in February 2023 only and three among three hotspot IUs in June 2023.

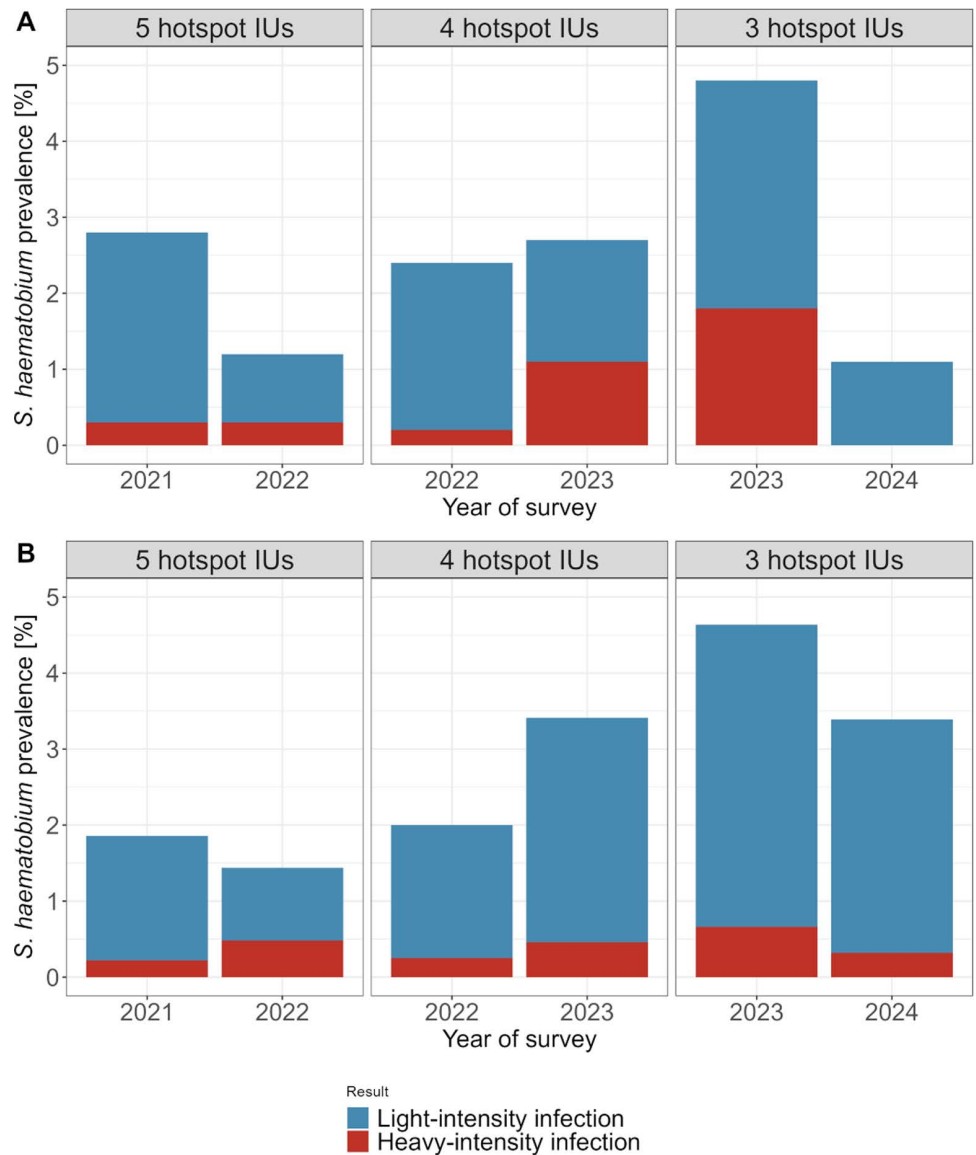

**Fig 4.** ***Schistosoma haematobium*** **prevalence in hotspot implementation units (IUs) in annual school-based surveys (A) and household-based surveys (B) of the SchistoBreak study in Pemba, Tanzania, from 2021 to 2024.** The figure shows the *S. haematobium* prevalence in hotspot IUs in the surveys before and after implementation of a comprehensive integrated intervention package. The number of hotspot IUs changed annually in line with the SchistoBreak study design.

one year of multidisciplinary interventions in these IUs, 1.2% (9/777) of the participating children were *S. haematobium*-positive and 0.3% (2/777) had a heavy-intensity infection. In the newly classified four hotspot IUs in 2022, 2.4% (10/424) of the participating children in the school-based survey had positive results and 0.2% (1/424) had a heavy-intensity infection. After one year of multidisciplinary interventions in the four hotspot IUs, the prevalence increased to 2.7% (12/444) and 1.1% (5/444) of participants had heavy-intensity infections. In the three newly classified hotspot IUs in 2023, 4.8% (19/397) of the children in the school-based survey were infected with *S. haematobium* and 1.8% (7/397) of the children had heavy-intensity infections. In the same schools, 1.1% (5/457) of the children in the school-based survey tested egg-positive in 2024 and 0% (0/457) of the children had a heavy-intensity infection.

In 2021, 1.9% (17/916) of the individuals participating in the household-based survey in the five IUs considered hotspots tested *S. haematobium*-positive (Fig 4B), and 6.9% (63/916) of the individuals tested positive for microhematuria (S1 Text and S1B Fig). Of the 916 participants, two individuals (0.2%) were diagnosed with a heavy-intensity infection. After one year of multidisciplinary interventions in these IUs, 1.4% (15/1044) of the participating individuals were infected with *S. haematobium* and 0.5% (5/1044) had a heavy-intensity infection. In the newly classified four hotspot IUs in 2022, 2.0% (16/800) of the participating individuals in the household-based survey tested egg-positive and 0.2% (2/800) had a heavy-intensity infection. After one year of multidisciplinary interventions in the four hotspot IUs, the prevalence increased to 3.4% (30/879) and 0.5% (4/879) of participants had heavy-intensity infections. In the three newly classified hotspot IUs in 2023, 4.6% (28/603) of the participants in the household-based survey were positive and 0.7% (4/603) had a heavy-intensity infection. In the same IUs, 3.4% (21/619) of the participants were infected in 2024 and 0.3% (2/619) had a heavy-intensity infection.

**Risk factors for *Schistosoma haematobium* infection in the SchistoBreak study area**

During the annual cross-sectional household-based surveys from 2022 to 2024, a total of 11576 individuals were included in the surveys, of whom 11518 community members were tested for *S. haematobium* infection using urine filtration. Of those, a total of 117 (1.0%) individuals from 15 IUs were infected with *S. haematobium* (Fig 5A). There were 5 IUs where no *S. haematobium* case was detected. Of the 117 individuals infected with *S. haematobium*, 38 (32.5%) resided within an area of moderate to high density of water bodies where *Bulinus* were detected (Fig 5B). Of the 11518 individuals assessed in the household-based surveys from 2022 to 2024, 1379 (12.0%) individuals had a very high economic score, 2035 (17.7%) had a high economic score, 2519 (21.9%) had a moderate economic score, 3072 (26.7%) had a low economic score, and 2513 (21.8%) had a very low economic score (Fig 5C). Of the 117 *S. haematobium*-infected individuals, 6 (5.1%) individuals had a very high economic score, 14 (12.0%) had a high economic score, 23 (19.7%) had a moderate economic score, 33 (28.2%) had a low economic score, and 41 (35.0%) had a very low economic score (Fig 5D). Of the 117 *S. haematobium*-infected individuals, 14 (12.0%) resided between 1 km and 2 km away from the next road, 18 (15.4%) resided between 500 m and 1 km away from the nearest road, 21 (18.0%) resided between 200 m and 500 m away from the next road, 29 (24.8%) resided between 50 m and 200 m away from the next road, and 35 (30.0%) resided <50 m away from the next road (Fig 5D).

From the household-based surveys in 2022, 2023, and 2024, data from 11518 individuals were included in a random effects model (Fig 6). The standardized kernel density of water bodies with *Bulinus* presence was significantly associated with *S. haematobium* infections (OR: 2.3, 95% CI: 1.6-3.4), indicating that for each one standard deviation increase in the density, the odds of infection increased 2.3 times. The odds of a *S. haematobium* infection were significantly higher for individuals with a low (OR: 2.5; 95% CI: 1.0-6.2), and very low economic score (OR: 4.1; 95% CI: 1.7-9.9), in comparison with individuals living in a household with a very high economic score.

A long distance from the house to a road was also a significant risk factor for a *S. haematobium* infection. Individuals living 500 m – 1 km away from the road (OR: 2.9; 95% CI: 1.5-5.5) and individuals living 1 – 2 km away from the road (OR: 4.7; 95% CI: 2.1-10.6) had higher odds of infection, compared with individuals living <50m away from a road.

Categorizing the 20 IUs by quartiles based on their mean chance of *S. haematobium* infection revealed that the first quartile, representing the highest infection chance, predominantly included IUs that were considered hotspots in three out of four years, specifically securing ranks 1, 2, and 5 (see Fig 7A and 7B). The IU, which was considered a hotspot in two years fell into the second quartile at rank 8. The IUs that were a hotspot in only one year were distributed across the first, second, and third quartiles, occupying ranks 3, 4, 6, 10, 16, and 17. The ten IUs that were never considered a hotspot were located mainly in the third and fourth quartiles, with ranks spanning 7, 9, 11–15, 18, 19, and 20, indicating a lower mean chance of infection.

**Fig 5. Characteristics of households participating in the SchistoBreak study in Pemba, Tanzania.** Location of study site in Pemba (green part in overview map) and location of households with a *Schistosoma haematobium* infected inhabitant (A), *S. haematobium*-positive households and kernel density of water bodies with *Bulinus* presence (B), household-economic status (C), and economic status and distance to roads of *S. haematobium*-positive households (D), assessed in community-based cross-sectional surveys from 2022 to 2024 in the SchistoBreak study area in Pemba, Tanzania. The image base map (United Republic of Tanzania – Subnational administrative boundaries) was downloaded from the UN Office for the Coordination of Humanitarian Affairs (OCHA) services (https://data.humdata.org/dataset/cod-ab-tza). The data source is: Tanzania National Bureau of Statistics/UN OCHA ROSA. The data are published under the following license: Creative Commons Attribution for Intergovernmental Organizations (CC BY-IGO; (https://creativecommons.org/licenses/by/3.0/igo/legal code)). Additionally, we received written permission to use and adapt the data from OCHA. The road data were retrieved from the Humanitarian Data Exchange Platform (https://data.humdata.org/dataset/hotosm_tza_roads), published under the Open Database License (ODC-ODbL).

## Discussion

The SchistoBreak study is among the first research projects that employed integrated multidisciplinary intervention packages for schistosomiasis elimination over multiple years, where interventions were targeted and adapted to the local micro-epidemiology, based on pre-selected prevalence thresholds. Our results show that the adaptive intervention

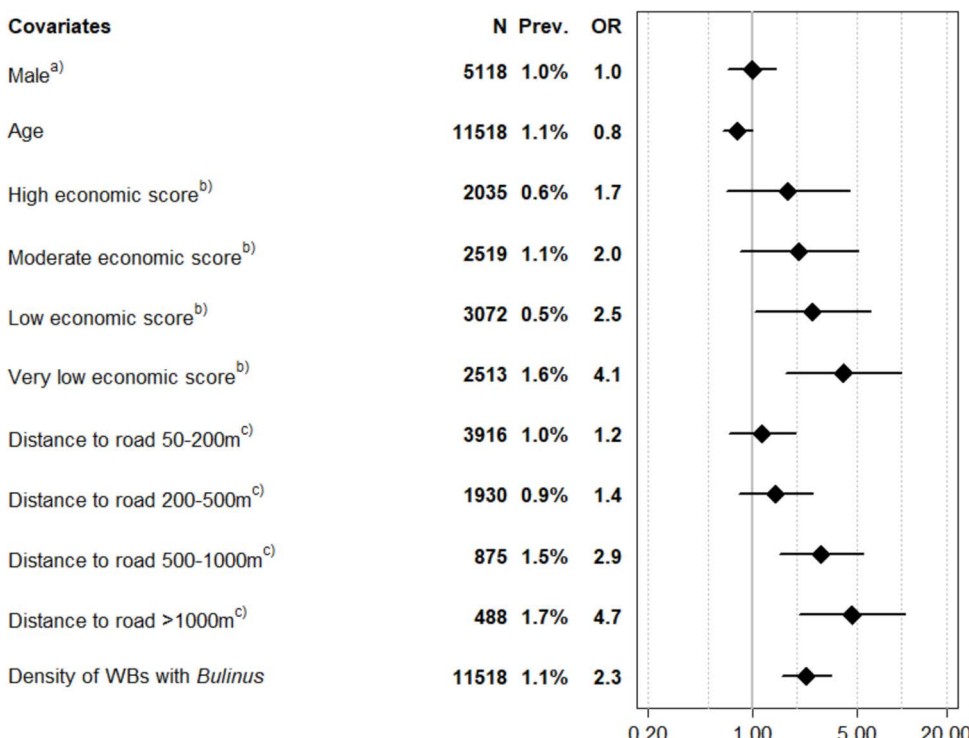

**Fig 6. Risk factors for *Schistosoma haematobium* infection.** Risk factors for *S. haematobium* infection in the household-based surveys of the Schis-toBreak study in Pemba, Tanzania, in 2022, 2023, and 2024. Prev. = *S. haematobium* prevalence, OR = Odds Ratio, WB = water body. Reference groups: a = Female, b = Very high economic score, c = Distance to road 0-50 m.

approach maintained the very low overall *S. haematobium* prevalence of ~1% (range: 0.8%-1.2%) in the schools and communities of the study area. Since heavy-intensity infections across the study area remained below 0.3% throughout all surveys, also the elimination of schistosomiasis as a public health problem was sustained. Yet, microhematuria (without trace) was detected in ~2% of students and in 3–6% of community members each year, pointing to residual morbidity due to urogenital schistosomiasis or microhematuria due to other causes, such as menstruation, urinary tract infections or bladder-stones [27, 35, 38]. Noteworthy, the adaptive intervention approach did not result in the interruption of transmission within the 4 study years.

The implementation of a combined intervention package consisting of at least annual MDA in schools and communities, repeated focal snail control and behavior change activities in hotspots, led to a decrease in the number of hotspot IUs across the first two intervention periods (5 in 2021, 4 in 2022 and 3 in 2023), while an increase was observed after the last intervention period (5 in 2024). Certain hotspot IUs remained hotspots despite the intense interventions and others became or rebounded to a hotspot IU once the hotspot intervention package ceased and they were exposed to the targeted surveillance-response approach implemented in low-prevalence areas [19, 30]. Looking at the individual hotspot IU sets per study period, there was a decreasing trend in the *S. haematobium* prevalence from 2021 to 2022, and from 2023 to 2024. From 2022 to 2023, the prevalence increased, mainly due to a large increase from 2.5% to 11.1% in one single IU. Of note, this specific IU remained a hotspot IU also in the following surveys, despite the intense interventions. Our results indicate that the comprehensive integrated intervention package helped to further reduce the overall *S. haematobium* prevalence in hotspot areas. However, the adaption to targeted surveillance-response measures and the subsequent rebound of infections in several hotspot IUs suggests that in hotspot areas, combined interventions need to be applied for extended periods to

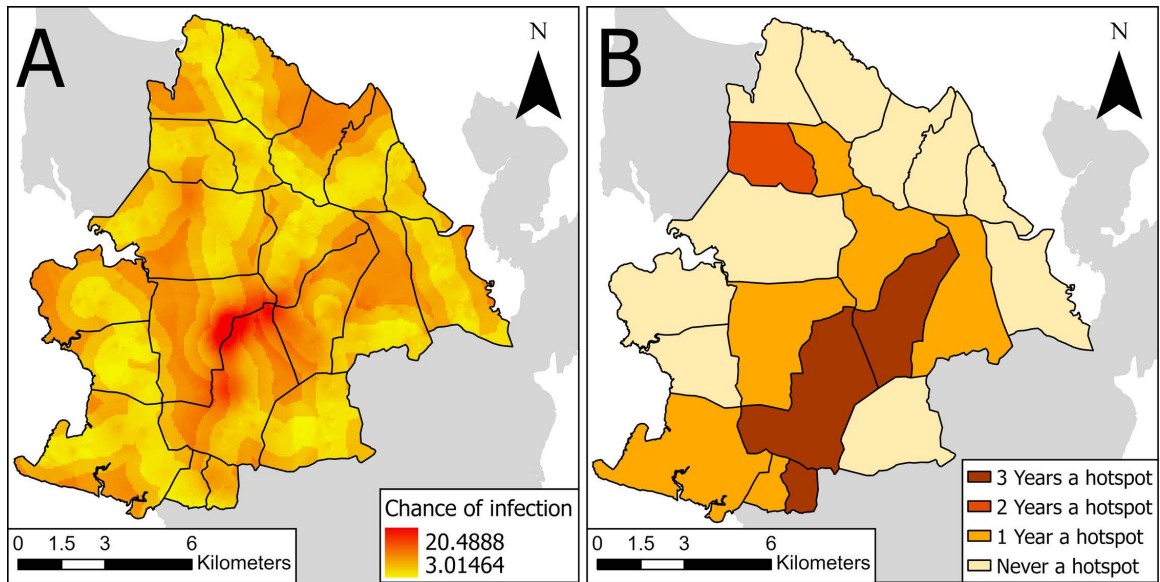

**Fig 7. Chance of infection with *Schistosoma haematobium*.** Chance of infection with *Schistosoma haematobium* in the north of Pemba (A) and implementation units considered hotspots based on *S. haematobium* prevalence in school and household-based surveys during the SchistoBreak study (B) in Pemba, Tanzania, from 2021 to 2024. The image base map (United Republic of Tanzania – Subnational administrative boundaries) was downloaded from the UN Office for the Coordination of Humanitarian Affairs (OCHA) services (https://data.humdata.org/dataset/cod-ab-tza). The data source is: Tanzania National Bureau of Statistics/UN OCHA ROSA. The data are published under the following license: Creative Commons Attribution for Intergovernmental Organizations (CC BY-IGO; (https://creativecommons.org/licenses/by/3.0/igo/legal code)). Additionally, we received written permission to use and adapt the data from OCHA.

break transmission, and that stopping MDA and other interventions can result in a quick recurrence. Our results are in line with a previous study from Zanzibar that reported a pronounced recrudescence in the *S. haematobium* prevalence, and particularly in hotspot areas, after a one-year MDA gap [27]. Also a study from China showed a recrudescence of *Schistosoma* infections after large-scale administration of praziquantel was stopped [39] and mathematical models indicate that the prevalence in high-risk areas likely rebounds once interventions cease [40].

However, one also has to note that the binary classification of hotspot and low-prevalence IUs as applied in our study is likely prone to random variation, particularly since our prevalence threshold was set very low and a few more or less positive participants may have caused a change in the classification. Moreover, when assigning interventions based on prevalence thresholds, it is important to be aware that the diagnostic methods used to determine the prevalence are imperfect and, when insensitive methods such as a single urine filtration are applied, the true prevalence is very likely underestimated [23]. Consequently, there is a risk for ceasing interventions such as MDA too early, resulting in recrudescence. Ideally, such stop-MDA decisions should not only be based on current prevalence data and thresholds, but also consider historical and long-term data, local knowledge, and environmental and social factors. Highly sensitive and specific tests should be applied to determine the prevalence on which the threshold and, hence, intervention implementation are based [41]. Moreover, mathematical models may be used to predict adequate prevalence thresholds and/or the occurrence of hotspot areas and help to assign adequate interventions [41–43]. Knowing the underlying factors of why hotspots are hotspots would support the prediction of such areas.

In our study, *S. haematobium* infections were significantly associated with: i) the density of water bodies containing *Bulinus* in close proximity to the infected individual's household, ii) a low economic status, and iii) the infected individual's households located far away from the next road. Our results are in line with studies from Cameroon, Ethiopia, and

Zimbabwe, where the proximity of communities to freshwater bodies harboring the intermediate host snail, and residence in a rural area were significant risk factors for *Schistosoma* infections [44–46]. A significant association between *Schistosoma* infections and low socioeconomic status was reported in a recent systematic review summarizing findings from different countries, including China, Côte d'Ivoire, and Sudan [47–50]. Using these factors in mathematical models may support the prediction of hotspots and the targeted assignment of interventions.

Clearly, achieving elimination in a formerly highly endemic area such as Pemba is not easy, especially if resources are limited. Intervention approaches have to be selected and adapted very carefully to the local micro-epidemiology to use resources adequately and yet avoid recrudescence. If zero autochthonous infections in humans is the goal in Pemba or elsewhere, it will be important to apply comprehensive integrated intervention packages with a high coverage across the island. But even then, one has to be aware of factors that jeopardize elimination. Reasons for the persistence of transmission and recurrence of infections are manifold. In Pemba, the large number of streams and ponds and the presence of the intermediate host snail *Bulinus* in many of these waterbodies provide an ideal and continuous transmission environment for *S. haematobium* [51, 52]. Snail control with niclosamide helped to reduce the number of *Bulinus* in our study area, but snails quickly returned and can thus contribute to maintaining transmission [32]. Animal reservoirs, such as cattle transmitting *S. bovis* and the potential for hybridization, should be considered as possible contributors to recrudescence [53, 54]. Behavior change interventions as applied in our hotspot IUs successfully improved the knowledge, attitude and practices of exposed children, but need to be complemented by improved access to water and sanitation to contribute successfully to elimination [34]. The MDA coverage in the schools and communities may be high as in our study, but will not reach all infected individuals, either due to insufficient coverage, non-attendance, or non-compliance [55–58]. Moreover, due to an imperfect praziquantel efficacy [59], infected individuals who participate in the MDA may not be fully cured and continue to excrete viable eggs that can perpetuate transmission. Additionally, in our study, the surveillance-response approach that was applied in low-prevalence IUs had only a moderate sensitivity and several infected individuals may not have been treated. Since people are mobile and may move within and between the Zanzibar islands and abroad, transmission may be reintroduced in areas free of schistosomiasis. Many of these factors are not specific for Pemba, but may consist a barrier to elimination also in other settings.

Hence, while continued MDA and additional interventions can help to reduce further and maintain a relatively low prevalence level in (former) hotspot areas, the risk for transmission and infection and hence a rebound in prevalence will remain as long as health literacy is low and poverty persists. Transmission and infection will continue as long as people do not understand the transmission routes and health consequences of schistosomiasis, and do not have easy access to infrastructure that allows the washing of clothes and dishes and showering with clean water at home, and to sanitary facilities including improved toilets at home and in proximity to natural open waterbodies. Reducing poverty, investing in WASH infrastructure and elevating the health literacy and socio-economic standard of people in areas endemic for schistosomiasis will remain vital to achieve the elimination goals set by WHO for 2030 and improve global health equity.

## Supporting information

**S1 Text. Microhematuria prevalence in hotspot implementation units.**
(PDF)

**S1 Fig. Microhematuria prevalence in hotspot implementation units.**
(TIF)

**S2 STROBE Checklist.**
(PDF)

**S1 Data   School data.**
(XLSX)

**S2 Data   Community data.**
(XLSX)

## Acknowledgments

We are very grateful to the several thousands of children and adults who participated in the SchistoBreak study and provided us with health-related information and urine samples. We thank all the principals of primary and Islamic schools, health facility staff and shehas of our study area for their great collaboration and support. We acknowledge WHO, Unlimit Health and the NTD team of the Zanzibar Ministry of Health for implementing regular MDAs in Zanzibar. Finally, we thank Bayer AG for donating the niclosamide for snail control interventions and conducting regular quality checks.

## Author contributions

**Conceptualization:** Lydia Trippler, Said Mohammed Ali, Jan Hattendorf, Stefanie Knopp.

**Data curation:** Lydia Trippler, Naomi Chi Ndum, Stefanie Knopp.

**Formal analysis:** Lydia Trippler, Jan Hattendorf, Stefanie Knopp.

**Funding acquisition:** Stefanie Knopp.

**Investigation:** Lydia Trippler, Said Mohammed Ali, Mohammed Nassor Ali, Ulfat Amour Mohammed, Khamis Rashid Suleiman, Saleh Juma, Stefanie Knopp.

**Methodology:** Lydia Trippler, Jan Hattendorf, Stefanie Knopp.

**Project administration:** Lydia Trippler, Said Mohammed Ali, Saleh Juma, Shaali Makame Ame, Fatma Kabole, Stefanie Knopp.

**Resources:** Said Mohammed Ali, Stefanie Knopp.

**Supervision:** Lydia Trippler, Said Mohammed Ali, Stefanie Knopp.

**Validation:** Said Mohammed Ali, Mohammed Nassor Ali, Ulfat Amour Mohammed, Khamis Rashid Suleiman, Saleh Juma, Shaali Makame Ame, Fatma Kabole.

**Visualization:** Lydia Trippler.

**Writing – original draft:** Lydia Trippler, Stefanie Knopp.

**Writing – review & editing:** Lydia Trippler, Naomi Chi Ndum, Jan Hattendorf, Stefanie Knopp.

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
