## [Decision Letter · Decision Letter 0]

5 Mar 2025

Response to Reviewers
Revised Manuscript with Track Changes
Manuscript

Shaden Kamhawi

co-Editor-in-Chief

Paul Brindley

co-Editor-in-Chief

**Additional Editor Comments (if provided):**
**Journal Requirements:**

At this stage, the following Authors/Authors require contributions: Lydia Trippler, Said Mohammed Ali, Mohammed Nassor Ali, Ulfat Amour Mohammed, Khamis Rashid Suleiman, Naomi Chi Ndum, Saleh Juma, Shaali Makame Ame, Fatma Kabole, Jan Hattendorf, and Stefanie Knopp. Please ensure that the full contributions of each author are acknowledged in the "Add/Edit/Remove Authors" section of our submission form.

**Reviewers' comments:**

**Key Review Criteria Required for Acceptance?**

**Methods:**

-Are the objectives of the study clearly articulated with a clear testable hypothesis stated?

-Is the study design appropriate to address the stated objectives?

-Is the population clearly described and appropriate for the hypothesis being tested?

-Is the sample size sufficient to ensure adequate power to address the hypothesis being tested?

-Were correct statistical analysis used to support conclusions?

-Are there concerns about ethical or regulatory requirements being met?

Reviewer #1: -Are the objectives of the study clearly articulated with a clear testable hypothesis stated?

- Yes

-Is the study design appropriate to address the stated objectives?

- Yes

-Is the population clearly described and appropriate for the hypothesis being tested?

- Yes

-Is the sample size sufficient to ensure adequate power to address the hypothesis being tested?

- yes

-Were correct statistical analysis used to support conclusions?

- Yes

-Are there concerns about ethical or regulatory requirements being met?

- No

Reviewer #2: (No Response)

Reviewer #3: (No Response)

**Results:**

-Does the analysis presented match the analysis plan?

-Are the results clearly and completely presented?

-Are the figures (Tables, Images) of sufficient quality for clarity?

Reviewer #1: -Does the analysis presented match the analysis plan?

- Yes, it does

-Are the results clearly and completely presented?

- Yes

-Are the figures (Tables, Images) of sufficient quality for clarity?

- Yes

Reviewer #2: (No Response)

Reviewer #3: (No Response)

**Conclusions:**

-Are the conclusions supported by the data presented?

-Are the limitations of analysis clearly described?

-Do the authors discuss how these data can be helpful to advance our understanding of the topic under study?

-Is public health relevance addressed?

Reviewer #1: -Are the conclusions supported by the data presented?

- Yes the conclusion is supported by the data, especially poverty is related to continual transmission of schistosomiasis

- Are the limitations of analysis clearly described?

- Yes, especially on the number of participants excluded in the analysis

- Do the authors discuss how these data can be helpful to advance our understanding of the topic under study?

- Yes, especially as we move towards 2030 goals of eliminating schistosomiasis

-Is public health relevance addressed?

- Yes, on the control of schistosomiasis in low prevalence areas and how integratied approach can help to maintain low prevalence

Reviewer #2: (No Response)

Reviewer #3: (No Response)

**Editorial and Data Presentation Modifications?**

Reviewer #1: None

Reviewer #2: (No Response)

Reviewer #3: (No Response)

**Summary and General Comments:**

Reviewer #1: The paper is good and described the impact of multiple intervention implemented to control schistosomiasis in areas with low prevalences. The work is relevant and deserve to be published. Furthermore, the research assess how poverty can contribute to continual transmission of this disease. To reach eradication goals, poverty must be addressed

Reviewer #2: This is a Research Article by Dr. Trippler and colleagues reporting on the SchistoBreak study performing a 4-year prospective adaptive intervention trial for Schistosoma haematobium elimination in low prevalence settings in Pemba, Tanzania. The authors applied a multi-disciplinary intervention (bi/annual MDA, snail control, behavioral change activities) to “hotspot” locations (defined as >3% school based prevalence or >2% household prevalence by urine microscopy) and surveillance / monitoring (with some directed test-and-treat) in non-hotspots based on annual repeated cross-sectional prevalence surveys across 20 locations. They found that while Schistosoma haematobium prevalence remained low with the multi-disciplinary intervention in hotspot locations, there was not elimination of transmission and prevalence remained overall the same. This is an important article reporting on a well conducted and impressive study on a pressing topic of elimination in the field of schistosomiasis. I would like to congratulate the authors and their team for completing this study. I hope my comments can be helpful and constructive.

Major comments:

--The definition of “hotspot” is highly variable in the literature of Schistosoma and beyond and the authors definition is different from others (e.g., WHO). For their purposes, it makes sense. However, it would be helpful to clarify in the Methods the justification for their definition, and also explicitly state that urine microscopy was used for the prevalence measurement. Furthermore, some additional Discussion on alternative hotspots definitions could be helpful (i.e., most are often used in the context of “control” rather than “elimination”). My final point is that this binary classification of hotspot when the prevalence threshold is so low is probably prone to random variation (i.e., a few positive participants may change the classification); when considering imperfect diagnostic sensitivity, drug efficacy, etc, it would be helpful to have some comments from the authors on the robustness of this classification, so that some fluctuation in number of hotspots is not over-interpreted.

--The study would benefit from additional discussion on other explanations for why this well-conducted and thought-out intervention did not eliminate transmission. For example, are participants traveling and getting infected elsewhere outside the implementation unit? Praziquantel efficacy is imperfect. Are we sure that all participants were treated in the prior MDA (given the design is a repeated cross-sectional and not the same cohort of individuals)? Did they all accept treatment? Animal reservoir? Size of transmission network?

Minor comments (most are just suggestions/optional):

--Abstract: In Principal findings, why not report 95% CI instead of the counts?

--Some discussion on generalizability of these findings (Pemba island) to other settings would be helpful.

--Were there differences in timing of the relationship between MDA and the prevalence survey that could explain some variation in year to year results?

--Were any participants tested with CAA or PCR? If not, would add this as a limitation (e.g., the prevalence threshold is based on a diagnostic test with imperfect sensitivity).

--Data visualization: Could add color legend to Figure 3. I found Figure 4 hard to understand, and perhaps some more explanation in the legend could help.

--Is there any other cause of the microhematuria in this population?

--Apologies if I missed this, but a figure plotting the resurgence of prevalence in nonhotspots given surveillance/monitoring would be quite helpful, as well as any unique features in those locations.

--The authors looks at risk factors for individual infection, but what about risk factors for being a hotspot IU vs non-hotspot IU?

Thank you for the opportunity to review this important work.

Reviewer #3: The authors provide a valuable contribution to the dearth of evidence for "integrated management" that the WHO is driving. It is incredibly noteworthy that the “integrated” approach did nothing. There is so little evidence to back this drive from the WHO across so many diseases, and it is nice to see an attempt to provide some evidence towards this.

My major qualm with this paper is that I am not sure I agree with your use of the term hotspot. You can have a persistent transmission hotspot where prevalence is still in your “low prevalence” threshold…your groupings are almost operating on different units of observation…And in some way I think this is why you see your hotspots coming back and going away in a way that to me fundamentally just seems quite random. There is no difference between these groups, the whole dataset is a low but consistent force of infection with some noise.

Figure 7 maps - it would help if these were zoomed out so that it is easier to see where your IUs fall in terms of the whole island.

Line numbers would make your feedback easier.

When you say you left out those with missing values for the PCA – how many did you start with and how many did you omit? Your sample sizes are not clear to me, I see you “aimed for 250 and 80 houses” but this isn’t a clear number of participants…I see now it is in your results section, I would find this easier to process in the methods section so I can appraise your methods more efficiently.

It would be nice to see your PCA output so that we can see the clustering of PCA1.

PLOS authors have the option to publish the peer review history of their article (what does this mean? ). If published, this will include your full peer review and any attached files.

**Do you want your identity to be public for this peer review?** For information about this choice, including consent withdrawal, please see our Privacy Policy .

Reviewer #1: No

Reviewer #2: No

Reviewer #3: No

**Figure resubmission:****Reproducibility:** To enhance the reproducibility of your results, we recommend that authors of applicable studies deposit laboratory protocols in protocols.io, where a protocol can be assigned its own identifier (DOI) such that it can be cited independently in the future. Additionally, PLOS ONE offers an option to publish peer-reviewed clinical study protocols. Read more information on sharing protocols at https://plos.org/protocols?utm_medium=editorial-email&utm_source=authorletters&utm_campaign=protocols

---

## [Editor Report · Decision Letter 1]

21 Apr 2025

Dear Dr Knopp,

We are pleased to inform you that your manuscript 'Adaptive integrated intervention approaches forschistosomiasis elimination in Pemba: a 4-year intervention study and focus on hotspots' has been provisionally accepted for publication in PLOS Neglected Tropical Diseases.

Best regards,

Jennifer A. Downs, M.D., Ph.D.

Academic Editor

Jong-Yil Chai

Section Editor

Shaden Kamhawi

co-Editor-in-Chief

Paul Brindley

co-Editor-in-Chief

---

## [Editor Report · Acceptance letter]

Dear Dr Knopp,

We are delighted to inform you that your manuscript, "Adaptive integrated intervention approaches for schistosomiasis elimination in Pemba: a 4-year intervention study and focus on hotspots," has been formally accepted for publication in PLOS Neglected Tropical Diseases.

Best regards,

Shaden Kamhawi

co-Editor-in-Chief

Paul Brindley

co-Editor-in-Chief
